



# An automated approach for developing geohazard inventories using news: Integrating NLP, machine learning, and mapping.

Aydoğan Avcıoğlu[1], Ogün Demir[2], Tolga Görüm[3]

[1]BRGM, 3 avenue Claude Guillemin, Orléans, 45060, France
[2]Nezahat Gökyiğit Botanic Garden, Biodiversity Information Department, İstanbul, Türkiye
[3]Eurasia Institute of Earth Sciences, Istanbul Technical University, İstanbul, Türkiye

*Correspondence to*: Aydoğan Avcıoğlu (a.avcioglu@brgm.fr)

**Abstract.** Spatiotemporal inventories of natural hazards are essential for comprehending the building of resilient societies; yet, restricted access to global inventories hinders the advancement of mitigation strategies. Consequently, we developed an approach that enhances the capability of online newspapers in the creation of natural hazard inventory by utilizing web scraping, natural language processing (NLP), clustering, and geolocation of textual data. Here, we use the online newspapers
from 1997 to 2023 in Türkiye to employ our approach. In the first stage, we retrieved 15,569 news by using our tr-news-scraper tool considering wildfire, flood, landslide, and sinkhole-related natural hazard news. Further, we utilized NLP preprocessing approaches to refine the raw texts obtained from newspaper sources, which were subsequently clustered into 4 natural hazard groups resulting in 3928 news. In the final stage of the approach, we developed a method, which geolocates the news using the Open Street Map (OSM) Nominatim tool, ending up with a total of 13940 natural hazard incidents
derived from news comprising multiple incidents across various locations. As a result, we mapped 9609 floods, 1834 wildfires, 1843 landslides, and 654 sinkhole formation incidents from online newspaper sources, showing spatiotemporally consistent distribution with existing literature. Consequently, we illustrated the potential of online newspaper articles in the development of natural hazard inventories with our approach from the web sources as text data to map by leveraging the capabilities of web scraping, NLP, and mapping techniques.

## 1 Introduction

Natural disasters are vital and direct threats to human life, ecosystems, and societies worldwide socio-economically, demanding ongoing innovation and development in the mapping, analysis, and monitoring of these events. The average annual economic loss due to natural hazards has been 34 billion US$ since 1900 according to The Emergency Management Database (CRED. 2023). According to the Sendai Framework, over 700 thousand people have lost their lives and >1.5
billion people have suffered from natural hazards a period between 2005 – 2015 (UNISDR 2015, p. 12). However, the assessment of damage and loss from different natural disasters might be underestimated. Because massive events such as large earthquakes and extreme wildfire events tend to trigger and cause subsequent disasters, such as landslides, debris



flows, flooding, soil erosion, etc. For instance, the Wenchuan earthquake, Mw 7.9 (USGS, 2008) triggered more than 60.000 landslides (Görüm et al., 2011) and its prolonged impact can influence even after years via landslide damming which is

recognized hazard due to the ability to release outbursts floods (Delaney and Evans, 2015; Fan et al., 2012, 2019; Peng and Zhang, 2012). Therefore, the lack of thorough evaluations and databases customized for specific hazard types causes temporal delays that hinder the understanding of dynamics of temporal and spatial probability and overall socio-economic and environmental losses of natural hazards. This obstacle results from the differences in data collection and monitoring practices among countries, each of which is subject to different legal frameworks.

Spatiotemporal archive inventorying is crucial for a better understanding of susceptibility, hazard, and risk assessment of natural hazards (Tanyaş et al., 2017, 2022; Loche et al., 2022; Gómez et al., 2023; Stein et al., 2024; Bhuyan et al., 2024). Also, these inventories can provide an objective base for resilience and preparedness for disaster risk reduction strategies (Jones et al., 2022). For instance, a well-known database, the Emergency Events Database (EM-DAT) operated under the Centre for Research on the Epidemiology of Disasters (CRED) provides a wide range of natural disaster inventory

(subgroups: Geophysical, Hydrological, Meteorological, Climatological, Biological, and Extra-terrestrial) with their corresponding casualties and economic loss (Guha et al. 2015). However, some studies (Froude and Petley, 2018; Görüm and Fidan, 2021; Haque et al., 2016; Stein et al., 2024) highlighted that EM-DAT lacks a thorough assessment of the natural disasters since it includes the events which resulted in the death of 10 or more people, 100 affected people, a declaration of a state of emergency, and a call for international assistance (Guha-Sapir et al. 2015). To overcome this constraint, some efforts

to establish global or national spatiotemporal natural hazard geo-databases are made, particularly for fatalities, utilizing systematic metadata search techniques that obtained from news articles, media sources, and national archives (Froude and Petley, 2018; Görüm and Fidan, 2021; Haque et al., 2016; Kirschbaum et al., 2015; Kirschbaum et al., 2010; Petley, 2012; Taylor et al., 2015). Beyond its value of comprehending natural hazards within the spatiotemporal inventories, these geo-databases frequently necessitate and rely on a workforce for labor-intensive tasks, such as compiling, gathering, and

analyzing the data in the creation of inventories.

Over the last two decades, there has been a noticeable advancement in the integration of artificial intelligence, namely machine learning (ML), deep learning (DL), and natural language processing (NLP), to create automated or semi-automated approaches for detection of natural disasters (Meena et al., 2022), monitoring (Restrepo-Estrada et al., 2018), early warning systems (Kitazawa and Hale, 2021), prediction (Fang et al., 2023), as well as the compilation and database

generation pertaining to natural disasters. However, the spatiotemporal data gathered by government and private databases are usually restricted or owned by private enterprises for profit (Lai et al., 2022). The internet sources like online newspapers and social media have been widely used to overcome this limitation by applying ML and NLP tools. For instance, Sodoge et al (2023) proposed an approach for automatization of drought impacts and creation spatiotemporal database based on newspaper articles in Germany between 2001 - 2021 using lasso logistic regression for impact detection and named entity

recognition in location identification. The spatiotemporal distribution of historical floods and storms was extracted using online newspapers on the United States-country scale by employing a hybrid named entity recognition model (Lai et al.,





2022). On the other hand, social media, particularly X (formerly known as Twitter), enabled the researcher to assess spatiotemporal patterns and to create a database of natural disasters by using data crawling methods (Franceschini et al., 2024). For instance, Hickey et al. (2024) tracked variations in the social reaction of geo-tagged Twitter posts during the 2018

2018 eruption of 18 Kīlauea, Hawaii, and found the reflective patterns of volcanic activities in the posts using sentiment analysis and ML tools.

Despite the wide application spectrum of NLP, ML, and DL tools in the creation of inventories or databases and assessment of natural disaster research, the studies mainly focus on single web sources and natural disasters such as drought (Madruga De Brito et al., 2020; Sodoge, Kuhlicke, Mahecha, et al., 2023), landslides (Battistini et al., 2013), flood (Liu et

al., 2020), typhoon (Kitazawa and Hale, 2021). Therefore, we developed an integrated method that retrieves, classifies, and geolocates multiple natural disasters; landslide, flood, wildfire, and sinkhole formation using web gazette sources in Türkiye between 1997 and 2023. We chose Türkiye as our research focus due to its proneness to natural disasters, leading to annual casualties (Görüm and Fidan, 2021) and socio-economic losses. Even though Türkiye highly suffers from earthquakes, we exclude the earthquake from our approach since geotagging problems due to the epicenter and news reporting distance

(Battistini et al., 2013) and also the international and national services provide freely available near-real-time spatial data of earthquake distribution.

The key goal of this research, therefore, is to develop a semi-automated approach for building spatiotemporal inventory and maps of natural hazards in Türkiye, such as sinkhole formation, wildfires, floods, and landslides. Our research focuses on creating a system that can parse newspaper articles about natural hazards from internet sources, classify the news

automatically according to the type of hazard, extract pertinent spatial coordinates and times of occurrence, and then map and store the collected natural disaster hazard data.

## 2 Methods and Data

To accomplish our targets, the general concept of the proposed approach includes five integrated main steps: (1) data collection from newspaper websites using a web scraper tool that we developed, (2) NLP preprocessing which cleans

data and extracts locations with named entity recognition (NER), (3) modeling; non-negative metric factorization for topic modeling, (4) geolocator which we developed an algorithm using Nominatim, (5) final inventory mapping.





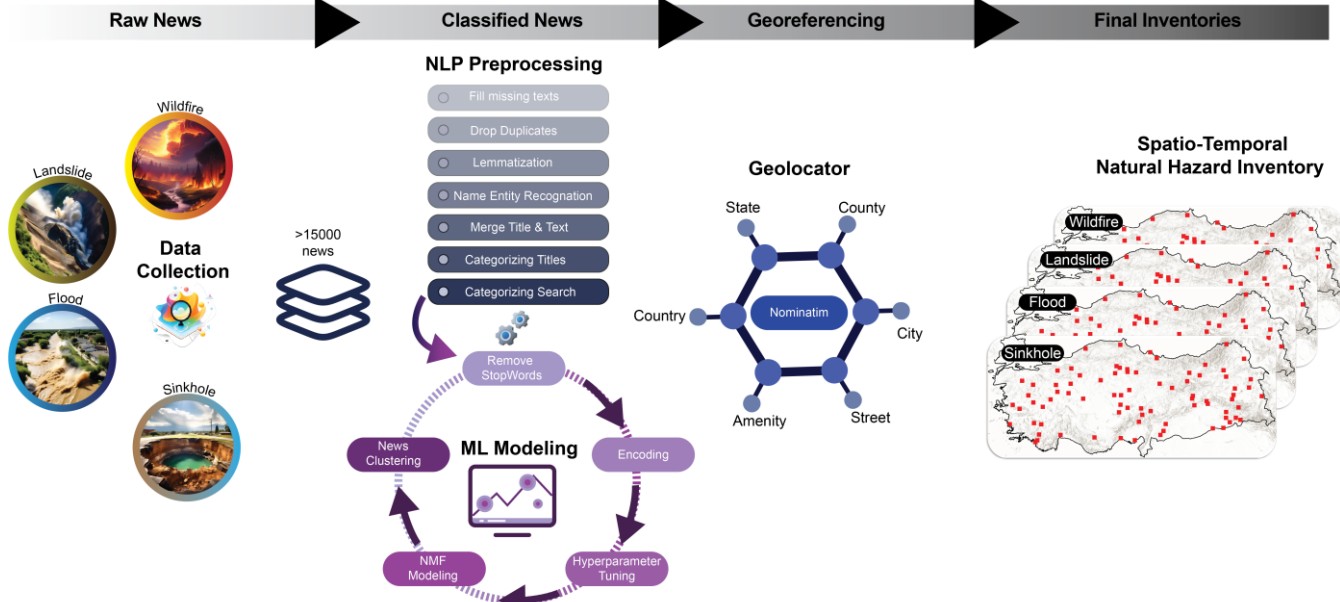

**Figure 1: The systematic flowchart of the methodology summarizing the steps employed from the data gathering to the final inventory mapping stages. The illustration of natural hazards by photos in the Raw News section of the figure were created with the assistance of OpenAI's DALL·E model.**


## 2.1. Data collection

We developed a Python tool "tr-news-scraper" (Demir and Avcioglu, 2024), to retrieve natural disaster-related news from newspaper websites. However, it's a flexible library that enables parsing any news by feeding some keywords as input. This application fetches HTML content from predefined websites of news by using the requests library. We targeted the wide-established national newspaper agencies that have been publishing for at least 10 years; Sabah, Milliyet, Hürriyet, CNN Türk, and Posta.

A list of keywords associated with natural hazards (Table 1) has been input into the tr-news-scraper. It fetches the URLs of newspaper articles which include each term by going through many pages of each news website. The scraper adds a delay between requests to avoid flooding the servers and getting blocked. The tool records metadata for every news article URL it fetches, containing the article's keyword and publishing date of the news. It also utilizes caching techniques to prevent retrieving the same URL twice.





**Table 1:** The keywords associated with natural hazards in Turkish are used to fetch newspaper articles in Türkiye.

| Categories | Keywords in Turkish | Keywords in English |
|---|---|---|
| Wildfire | "orman yangını", "orman yandı", "çalılık yangını", "makilik yandı" | "the forest fire", "forest burned", " bushes burned", "maquis burned" |
| Flood | "sel", "sel meydana geldi", "taşkın oldu", "nehir taştı", "çamur aktı", "dere taştı" | "flood", "flood occurred", "there was a flood", "the river overflowed", "the mud flowed", "the creek overflowed" |
| Landslide | "heyelan", "toprak kayması", "kaya düştü", "kaya düşmesi", "toprak aktı" | "landslide", "earth slide", "the rock fell", "rockfall", "the earth flowed" |
| Sinkhole | "obruk meydana geldi", "obruk oluştu" | "a sinkhole occurred", "a sinkhole formed" |

Subsequently, the content of each news article is retrieved by the scraper tool after gathering all of the URLs. It takes advantage of the newspaper library to make article extraction and processing easier. The scraper gathers relevant data for every URL, including the title of the article, the author(s), the date of publication, the primary text content, related keywords, and any images that are accessible. Following extraction, the data is structured into a Pandas DataFrame (Supplementary Table 1) to ensure that it might be examined further. The scraper eliminates duplicate entries based on both textual content and URL to ensure data integrity. In essence, the tr-news-scraper tool retrieves news articles about natural hazards automatically from several websites, giving a large dataset for further analysis and classification.

## 2.2. NLP Preprocessing

The unrefined retrieved data from web sources, as stated above, typically lacks spatial coordinates and is unstructured. The context of news is transformed multiple times throughout the preprocessing stage before being analyzed. NLP preprocessing requires multiple steps to get text (i.e., news) input ready for additional modeling or analysis. In advance of analysis and modeling, Hickman et al., (2022) portrayed the following common steps for text preprocessing: text identification (i.e., tokenization), content removal (i.e., stop words and nonalphabetic characters removed), agglomeration of semantically related terms to reduce data sparsity and improve predictive power (i.e., lowercase conversion, misspelling correction, contraction/abbreviation expansion, and stem/lemmatized), and capturing more semantic information (i.e., handling negation).

We have followed the steps shown in Figure 1 to complete the cleaning and preprocessing of the unrefined data parsed from web sources to get ready for further analyses. Initially, we filled in the missing contents (the main body of the news) with the meta description text which summarizes the parsed news related to our target keywords. This step was necessary owing to contents where news articles contain titles but lack corresponding content during parsing procedures. The duplicates that originate from the same media sources yet differ in keywords have been removed from the retrieved dataset. We created a new column by merging the titles and contents of the individual columns we parsed, which better represented each event (such as a flood, landslide, etc.) or unrelated news concerning our target. Because titles provide succinct, event-



specific information that highlights the distinctive features such as type, location, and date of natural hazards. Subsequently, we utilized the lowercase conversion which is a standard application in NLP (Hickman et al., 2022) to new merged content which includes titles and the main body of the news. Also, the removal of punctuation of content was a necessary step in achieving our objective of cleaning noisy text by using specific filters (noun, adjective, adverb, and verb), thus enabling the elimination of conjunctions, punctuation marks, articles, etc. After that, in order to improve the coherence and consistency

of our text analysis, we lemmatized our content using the TrSpaCy pipeline (Altinok, 2023) to break words down to their most basic forms utilizing linguistic processing approaches. Lemmatization is a step that helps to standardize terminology and makes semantic analysis across the corpus more accurate.

When identifying the stopwords for the natural disaster hazard news articles, it is necessary to compile a detailed list of commonly used terms that are unrelated to the incident. By removing noise and unimportant information from the

original stopword compilation, content (i.e., type, date, location of event) directly related to the disaster of interest can be identified more accurately. We may fine-tune our search criteria and improve the accuracy of our data retrieval efforts by deliberately eliminating irrelevant terms from our analysis.

Vectorization is another essential step for text classification in NLP. We utilized the Term Frequency (TF) - Inverse Document Frequency (IDF) technique (TF-IDF) which is a widely used statistical method in NLP and feature extraction. We

perform the TfidfVectorizer using the scikit-learn (Pedregosa et al., 2011) in scaling the words. TF is the number of times it appears in a document in relation to the total number of words in that document, Eq 1.:

$$TF = \frac{\text{number of times the term appears in the document}}{\text{total number of terms in the document}} \qquad (1)$$

A term's IDF shows how many documents in the corpus contain that term. Words that are specific to a limited subset of papers (for example, technical jargon terms) are given a greater relevance value than words that are used in all

publications (a, the, and), Eq 2.:

$$IDF = log\left(\frac{\text{number of documents in the corpus}}{\text{number of documents in the corpus contain the term}}\right) \qquad (2)$$

Then TF-IDF can be calculated by multiplying TF and IDF scores:

$$TF\text{-}IDF = TF * IDF \qquad (3)$$

The n-grams, a hyperparameter in the TfidfVectorizer, were applied as "ngram_range = (1,2)" for unigrams and diagrams capturing both single words and pairs of words within the specified range to better represent nuanced text data.

Named Entity Recognition (NER) is the process of locating specific words or phrases, so-called "entities", in a document and categorizing them into groups like people, places, or events. It helps to grasp the context and meaning of the text, which is important for a variety of natural language processing applications, such as sentiment analysis and information

extraction. We used the NER component of the TrSpaCy which is the first spaCy model trained for the Turkish language produced by using diverse sources in the Turkish language: Wikipedia articles, crawling of e-commerce, and movie review



websites for different genres (Altinok, 2023). This model essentially includes a tokenizer, trainable lemmatizer, POS tagger, dependency parser, morphologizer, and NER pipelines.

## 2.3 Modeling

Once the extracted unrefined data has been successfully preprocessed, the next step is to feed these refined contents into several models for additional processing and prediction. First, the most relevant keywords related to different natural disasters are identified using the Nonnegative Matrix Factorization (NMF) technique, which was first pioneered by Paatero and Tapper (1994, 1997) as well as Lee and Seung (1999, 2001).

2.3.1 Nonnegative matrix factorization (NMF)

NMF refers to a set of linear algebra and multivariate analysis techniques where a matrix $X$ is divided into two matrices, $W$ and $H$, each of which only has non-negative elements by minimizing the distance $d$ between $X$ and the product of WH. The most widely used distance metric is the squared Frobenius norm, which is a simple matrix application adaption of the Euclidean norm (Lee and Seung, 1999), Eq. 4:

$$d_{\mathrm{Fro}}(X,Y) = \frac{1}{2} \|X - Y\|_{Fro}^{2} = \frac{1}{2}\sum_{i,j}(X_{ij} - Y_{ij})^{2} \qquad (4)$$

This approach, which is an unsupervised learning technique, reduces the dimensionality of data into spaces of fewer dimensions. We used the NMF model as an additional step for data filtration which is initially categorized into different types of natural disasters in the preprocessing section. This model identifies and finds efficiently the different clusters of natural disasters. Because the original data set parsed through our tr-news-scraper method contains irrelevant and noisy information that originated from search engine optimization (SEO) practices of the news websites. This method incorporates
news information that can increase news visibility to search engines and their user base.

NMF model suggests clusters that are either natural hazards or not by using the parsed and preprocessed news content as input data. Further, we expand this division to encompass different categories of natural hazards by increasing the number of natural clusters within the NMF model that we seek to autonomously detect. As mentioned above, the primary goal of the NMF model is to identify and differentiate the themes related to different natural hazards and others (i.e., not
related to natural hazards; health, politics, sport, etc.) that also need to be evaluated by further validation procedures. Therefore, we utilized the coherence score and expert-based evaluation to validate the results of the NMF model. The coherence score measures the semantic similarity between high-scoring words within each topic, evaluating how interpretable and meaningful the topics are (Rehurek and Sojka, 2010; Röder et al., 2015). To assess the coherence of the news topics produced by the NMF model, we employed the Coherence Model from the Gensim library (Rehurek and Sojka,
2010). We utilized $Cv$ as the coherence option in CoherenceModel. $Cv$ uses a sliding window approach, grouping the top words into a single set and employing an indirect validation metric that combines normalized pointwise mutual information (NPMI) with cosine similarity (Rehurek and Sojka, 2010; Röder et al., 2015; Syed & Spruit, 2017). Furthermore, true (i.e., incident) and false (i.e., not incident) tests are determined by authors based on textual content that compares with those



produced by the NMF model results to perform an expert-based evaluation. The classifications created by authors were
regarded as actual data. The accuracy of the NMF-generated categories was then evaluated, and their alignment with the
human-defined categories was used to determine whether they were correct, yielding the evaluation score. For this task, we
have chosen random 2000 news over 10593 total news within all-natural hazard news, and we have evaluated the final score
by reading and determining their "incidents" criteria (i.e., true and false). It is essential to highlight that if the news meets our
criteria—that is, provides a clear explanation of the occurrence of specific events with date and geographical attributes—we
have classified it as an "incident" in our inventories (Supplementary Table 2). Therefore, we have eliminated the news such
as regional or temporal reviews and repeated news.

### 2.4. Geolocator

In this study, we developed a geospatial data processing and localization method leveraging the locally hosted
Nominatim geocoding service. The primary objective of this method is to determine geographic locations from textual
locality descriptions accurately. Because the information structured in the RSS or Atom format does not have a native
geographical location; the news itself is not associated with any structured geo-location (Battistini et al., 2013). The process
is implemented in Python. Our method processes the textual locality descriptions and retrieves geolocation information. It
constructs and sends queries to the Nominatim API for each locality entry, parsing the results to identify geographic
components such as states/provinces, counties, cities, amenities, and streets. The method follows a systematic approach. It
begins by searching for states/provinces within the locality descriptions. Upon finding a match, it then sequentially searches
for counties, cities, amenities, and streets within the remaining locality descriptions. The implemented method efficiently
processes large datasets of locality descriptions, accurately identifying and organizing geographic components.

### 3 Results and Discussion

A total 15569 number of articles from 1997 to 2023 have been fetched from newspaper websites through our web
scraping tool "tr-news-scraper" by using the keywords listed in Table 1. The raw inventory includes 5510 floods, 4262
wildfires, 5255 landslides, and 542 sinkholes. Following the first filtering, which involves eliminating redundant, repeated,
and unnecessary news, a total of 4,000 news remain for the subsequent stages of NMF modeling and geolocalization (Table
2). Geohazard news from NMF is grouped into 2236 floods, 655 wildfires, 766 landslides, and 271 sinkholes, resulting in
3928 news remaining which include multiple locations. Nevertheless, following a thorough semantic analysis using TrSpaCy
(Altinok, 2023), we identified 13940 distinct locations (i.e., cities, counties, villages, etc.). As a result, we have determined
that these areas have at least 13940 geohazard incidents.






**Table 2: The analytical processes of different classes of news at various stages of the analysis from obtaining unrefined data to geolocalized news as an inventory.**

| Class | Flood | Wildfire | Landslide | Sinkhole | Total |
|---|---|---|---|---|---|
| Unrefined News | 5510 | 4262 | 5255 | 542 | 15569 |
| Filtered News | 4270 | 2123 | 3860 | 331 | 10593 |
| NMF Groups | 2236 | 655 | 766 | 271 | 3928 |
| Geolocalized Incidents | 9609 | 1834 | 1843 | 654 | 13940 |


Figure 2 shows the WordCloud of different natural hazards and Supplementary Table S2 summarizes the most important 20 words with their frequencies related to the NMF model results. As expected, the total number of words is highest in the flood news and lowest in the sinkhole, it is correlated with the number of news parsed. "Yangın" (fire) and "orman" (forest) are the two most commonly (3.28% and 2.59%, respectively) used terms about wildfires. Given how frequently "orman" appears in the news, it implies that we have compiled the news from forested areas which matches the wildfire criteria for this study. Because we haven't included news on fire occurrences involving buildings, homes, etc. Furthermore, bigram combinations of the most frequent words reveal that the term "orman yangını," which is translated as "wildfire," is the most often occurring noun group in NMF grouping analyses. This shows that urban fires—such as those that occur in buildings, homes, etc.—were eliminated from the inventories and analyses.



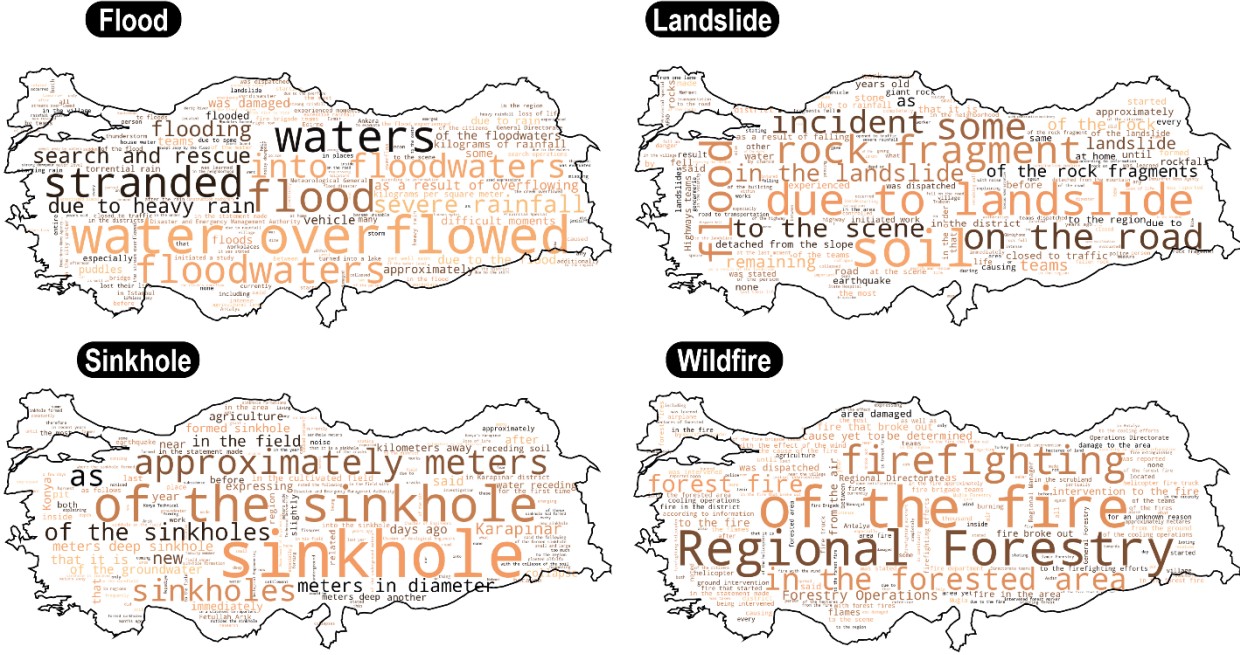


**Figure 2: The world clouds for different natural hazards emphasize the most frequently seen words in the filtered news.**

The phrases "sel" (flood), "su" (water), "yağış" (precipitation), and "sağanak" (downpour) are associated with flooding, indicating a strong correlation between heavy rainfall (the term "sağanak") and subsequent flooding incidents. As
indicated by the prominence of these phrases, which highlight the importance of water-related disasters and their direct relationship to precipitation patterns. Despite being documented as less frequent occurrences, sinkhole events can be identified by their characteristic phrases within the news. The term "metre" (meter) and the term "obruk" (sinkhole) are closely associated, it is possible that understanding the significance of sinkholes depends critically on their size or depth. These terms are frequently used, which emphasizes how crucial it is to measure and monitor sinkholes because they can
cause major disruption to infrastructure and public safety.

The terms connected to landslides are "kaya" (rock), "toprak" (soil), and "heyelan" (landslide). These keywords also emphasize the geological origin of the component of landslide events by indicating the role of rock and soil. It is clear to identify the type of landslide activity from the most frequent terms from the table that "düştü" (fell), "kopan" (broken off), and "parçala" (pieces of) point particularly rockfall activities. These terms may also point to the selective importance of
landslides, which are especially newsworthy because they directly affect people's lives, urban areas, and vital infrastructure. It is generally acknowledged that journalists often agreed on exclusive matters of public relevance (Harcup and O'Neill, 2017; Pita Costa et al., 2024). Therefore, the number of landslides in our inventory may primarily reflect landslides that impact humans because most landslides occur in remote mountainous areas or away from human infrastructure, etc., and are less essential than others, causing some socio-economic losses. It is important to emphasize that certain phrases may give





rise to confusion when it comes to grouping analysis. For instance, the term "yağış" (precipitation) is significant for both landslides and floods. Since most news reports highlight landslide incidents by mentioning predisposing elements like precipitation—which is also important for flood events, this could result in mis-clustering, primarily in landslide events.

The Non-negative Matrix Factorization (NMF) model was applied to the news dataset using a range of topic numbers from 2 to 20 components. The coherence score was calculated for each model configuration to evaluate the
coherence and interpretability of the generated topics. Supplementary Figure 1 shows that the coherence score generally increased with the number of topics, reaching its plateau at 4 components with a coherence score of 0.80. This suggests that the 4-topic model provides the most meaningful and coherent topics for the given news dataset. Beyond the 4 components, the coherence scores vary or decrease slightly, indicating that additional topics do not contribute significantly to the model's overall interpretability. The selected 4-topic model thus strikes an optimal balance between topic coherence and topic
interpretability, providing a robust representation of the underlying thematic structure of the news dataset. Also, our expert-based evaluation score, which we performed by 2000 randomly selected news over 10593, showed overall good consistency with a coherence score of 0.81 evaluation score. On the other hand, when it comes to evaluating each type of natural hazard, flood, wildfire, landslide, and sinkhole, they resulted in different scores, 0.84, 0.7, 0.85, 0.74, respectively. This evaluation is mainly based on the incident identification criteria. For instance, we have determined four major categories, leading to
misclassification in our incident identification. These are the news, that is categorized as "not incident" as shown by the 0 values in Supplementary Table S3. The first category is the "common words" with incidents like "a person lost her/his balance and fell while rock climbing" (in Turkish: "*kaya tırmanışı yaptığı sırada dengesini yitirerek düşen*"). The second category is the "review news" which essentially compiles multiple incidents over the course of time. The third category is "warning alerts" such as "AFAD issues forest fire warning for 6 provinces…" (in Turkish: "*AFAD'dan 6 il için orman
yangını uyarısı*…"). The last category is "misinterpretation" as given in the example: "… *They encountered a sinkhole that was 7 meters wide and 10 meters deep due to a meteorite fell.*" (in Turkish: "*... Meteorite düşmesinden kaynaklı 7 metre genişliğinde ve 10 metre derinliğindeki obruk ile karşılaştılar...*").

The temporal distribution of the hazards shows an increase after 2005 (Fig. 3). This result implies that internet sources became more widely available after 2005, which is in line with the increase reported by Gorum and Fidan (2021) for
fatal landslide cases in Türkiye.  Throughout the study period (1997–2023; Fig. 3), sinkholes and wildfires fluctuated, but the frequency of floods has shown an increase, particularly after 2016, when the annual number of events increased approximately from 400 to 1600 by 2023. Similarly, even though the landslide numbers showed an increase after 2016, the frequency of landslides (Fig. 3) has remarkably decreased (-60 % and -36 % last two years, respectively).  The European Forest Fire Information System (EFFIS) (San-Miguel-Ayanz et al., 2012) database, which records the largest burnt area in
Europe at 27848.33 hectares (https://forest-fire.emergency.copernicus.eu/reports-and-publications/annual-fire-reports, last accessed August 2024), suggests that although wildfire occurrences exhibited sporadic pattern, their peak occurrence in 2008 was consistent with our database (Fig. 3).





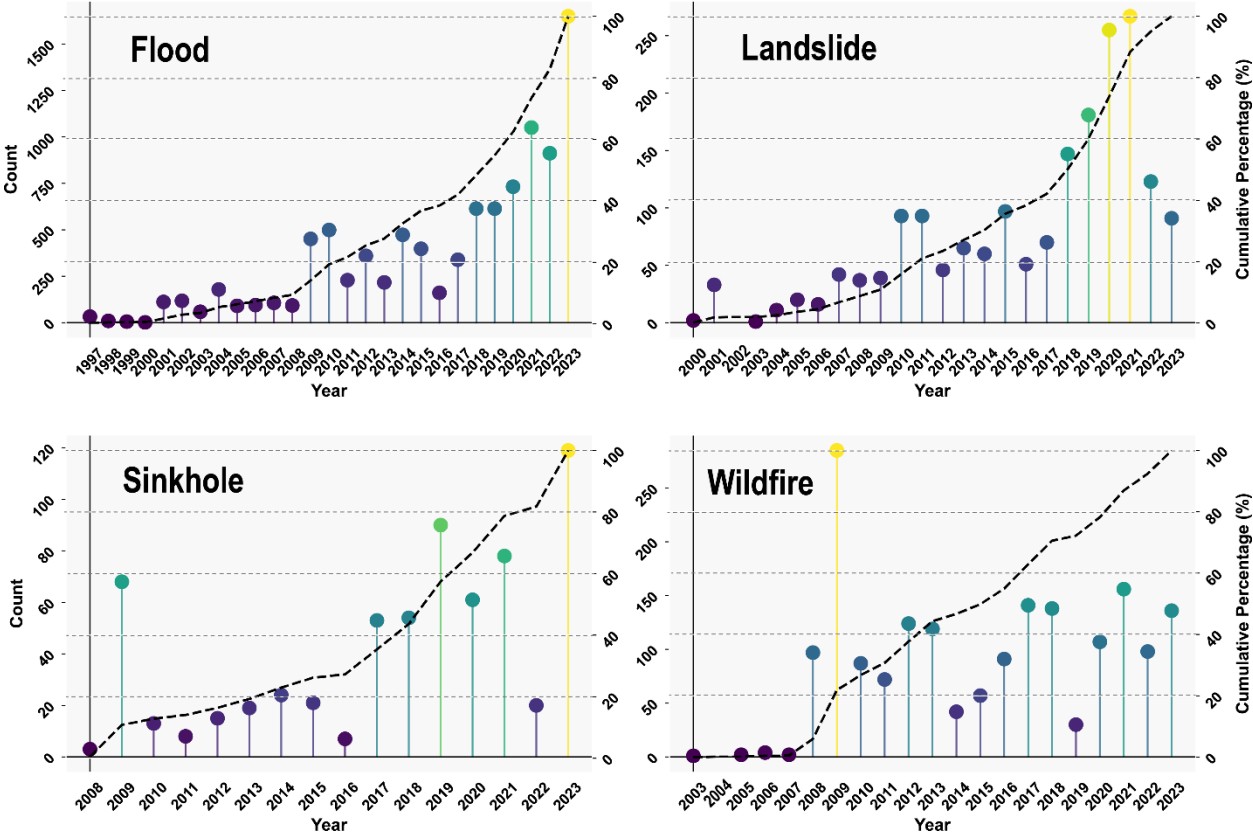

**Figure 3: The temporal distribution of the flood, landslide, sinkhole, and wildfire incidents that we mapped. The**
**beginning year varies for different natural hazards and the dashed line indicates cumulative incidents over the study**
**period as a secondary axis. \*Count refers to the number of incidents of natural hazards.**

Hu (2018) highlighted the significance of ambiguous connections between texts and locations, meaning that geo-text data
can contain both from and about locations (MacEachren et al., 2011). This problem implies that many location names have
turned prominence in the news, which can lead to georeferencing complications. With this problem in mind, after acquiring
the 3928 total clustered news (i.e., NMF Groups in Table 2), we used TrSpaCy (Altinok, 2023)—a geoparsing technique that
extracts explicit place names from implicit geo-text data (Gregory et al., 2015; Hu, 2018)—to acquire place names. To
further address the point raised by Hu (2018), the spatial filters also removed the names of other cities from the news,
leaving only one.
It should be taken into account that the number of natural hazards represents the minimum number of incidents that occurred
during the period in our analyses. The geohazards events we fetched from the online gazettes are newsworthy with several
aspects that cause economic losses in many ways; damage to critical infrastructures, urban areas, and agricultural activities.
Given that floods and wildfires have numerous effects on human life and are more frequently observed, there may be less





variation between actual occurrences and the incidents we record compared to landslides and sinkholes even if they cause
severe loss. The reason behind this is that the internet gazettes or online sources do not consider landslides that occur
remotely from metropolitan areas or vital infrastructure to be noteworthy.

### 3.1 Spatiotemporal implications variations on natural hazard inventories

Here, we demonstrate the spatiotemporal distribution of natural hazard inventories that have been examined using thorough
semantic analysis and Natural Language Processing (NLP) techniques at the national level in Türkiye and we compare
spatiotemporal consistencies of our inventories with existing literature.

**Figure 4: The spatial distribution of natural hazards over different administrative regions of Türkiye. The circular bar plots depict the percentages of events across various regions, accompanied by density maps illustrating spatial hotspots of natural hazards across different regions in Türkiye.**



Given the primary concentration of natural hazard occurrences (Fig. 4), Türkiye displays a particular spatiotemporal tendency to geohazards (Fig. 5). The spatial distribution of natural hazards reveals that flood events are relatively well distributed over the country to others. But the majority (54 %) of the events happened in the Marmara (29.4 %) and Black Sea (24.6 %) regions, less common in the Central Anatolia Plateau (Fig. 4), the driest region in Türkiye, especially around the Konya province (average ~400 mm/y precipitation). Some hotspots for flood events appear near big cities like Istanbul,

Ankara (the capital city), and İzmir (Fig. 4). This may suggest that large cities can readily get the attention of gazettes, even for minor incidents. Ankara, for instance, may appear as a hotspot, yet the Central Anatolia Region, which includes it, has had 6.7 % of all floods. On the other hand, Istanbul is not only the biggest city in the Türkiye which receives major attention from journalists, but also it is, particularly in north-faced basins, geographically part of the Black Sea region resulting in higher rainfall which potentially impacts the number of floods. Therefore, attention should be given to these inventories for

the real number and accuracy of the events. It is important to note that our inventory primarily captures the urban floods that mainly occur where there is a construction area in flood-prone areas (Brown et al. 2007; Mason et al. 2007) and poorly engineered flood control infrastructure (Gallegos et al. 2009; Ozdemir et al. 2013).

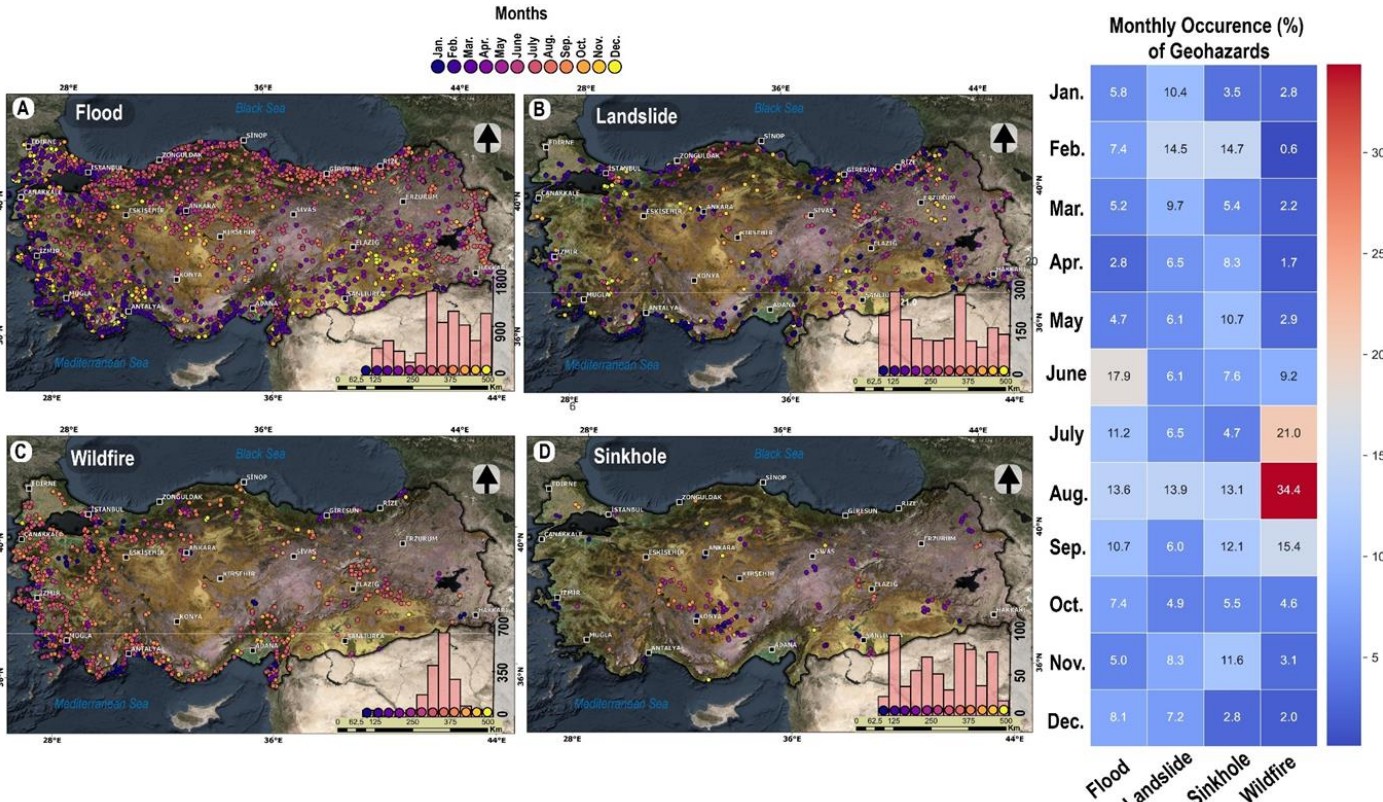

**Figure 5: The spatiotemporal distribution (monthly) maps of natural hazards, which are aligned with bar plots**

**showing the total number of months over the study period from January to December in the color gradient from blue**





**to yellow, respectively. The heat map chart demonstrates the monthly occurrence (in %) for each natural hazard group.**

The temporal distribution of the flood events also shows different patterns by region (Fig 5.). The summer season accounts

for 42.7 % of all floods in Türkiye, and this temporal pattern shifts to the winter season in cities along the Mediterranean coastline (Fig. 5). Additionally, the spatial distribution of flood events is consistent with the findings of earlier research (Haltas et al., 2021). For example, Koç et al. (2020) showed that fatalities and economic losses due to flood events in the Marmara and Black Sea regions where we identified the majority of the events (54 %).

The spatial distribution of landslides in our inventory shows that the Black Sea Region is the most susceptible area to

landslides by having the majority (42.9 %) of all events over Türkiye. The current literature indicates a comparable spatial distribution, with fatal landslides (Görüm and Fidan, 2021) and Turkish landslide inventory (Duman et al., 2011) predominantly concentrated in the Black Sea Region, especially in the eastern section. As indicated in the first section regarding the flood and landslide events being wrongly classified due to the common keywords (e.g, "yağış"), Figure 4 also portrays a supportive spatial relationship in the East Black Sea region where the higher slope and rainfall and the higher

flood and landslide events occur. This misclassification mainly affects landslide incidents since most of the news highlights meteorological conditions in the news such as "*landslide occurred as a result of prolonged precipitation.*" (in Turkish; "*uzun süren yağış sonucu heyelan meydana geldi.*"). On the other hand, whilst this study does not concentrate on triggering or predisposing factors of the natural hazards, it is important to note that the landslides where we obtained fewer occurrences; Marmara, Central Anatolia, and Southeast Anatolia regions are more likely to be associated with anthropogenic origin

activities such as mining, road cutting, and other related activities (Fidan and Görüm, 2020; Görüm and Fidan, 2021).

There has been a spatial accumulation of wildfire incidents in our inventory mainly along the Mediterranean coast from Adana to Çanakkale, (Fig. 4 and 5). Conversely, the vicinity of Istanbul exhibits a concentrated region for wildfire occurrences (Fig. 4), indicating increased frequency; these wildfires are notably characterized by a smaller magnitude (i.e., areal coverage) compared to the Mediterranean coast. It is noteworthy to highlight that the Mediterranean and Aegean

regions show a distinct tendency with their topography (Avcioglu et al., 2024), climatology (Tatli and Türkeş, 2014; Akbas, 2023), and predominant vegetation cover; *Pinus brutia* Ten. (kızılçam) (known as also Turkish pine) and *P. halepensis* Mill. (halep çamı) (Ekberzade et al., 2022), accounting for the majority of wildfire cases (~ 65%). Türkiye exhibits an evident seasonal pattern throughout the summer months, especially along the coastline that stretches from Marmara to the Mediterranean (Fig. 5). A recent study by Öztürk et al. (2024) identified significant wildfire areas attributed to lightning,

demonstrating spatial consistency with our inventory mapping, namely hotspots of wildfires over the Mediterranean and Aegean regions of Türkiye. On the other hand, an interesting finding points out that wildfire incidents occur during the winter and spring in the Eastern Black Sea Region (one of the less frequent regions). This might suggest the influence of the natural phenomenon "foehn winds" a type of dry, relatively warm downslope wind that occurs in the lee (downwind side) of




a mountain range which elevates wildfire risk as a potential driver and predisposing factor. This phenomenon also has been shown in the studies (Yetmen and Aytaç, 2017) highlighting the importance of foehn winds in the wildfire case in the Eastern Black Sea Region.

The sinkhole formation is the least common natural hazard in Türkiye compared to others in our inventory. In fact, literature has demonstrated that sinkholes are among the most significant natural hazards (Waltham and Fookes, 2003; Parise et al., 2008) because of their rapid and unexpected occurrences, which restrict the certainty of their spatiotemporal forecast (Newton, 1987). Consistent with the literature (Doğan and Yılmaz, 2011), Figure 4 clearly shows that sinkholes predominantly occur in the Central Anatolia Region with no particular temporal tendency (Fig. 5), specifically the Obruk Plateau (subregion) that surrounds the Konya province. In addition to this region, sinkholes have also been reported in the news as incidents from other regions. Although geological settings have made it less likely for sinkholes to occur in other regions, we maintained all incidents because our approach primarily depends on context truth (i.e. article within news) rather than geologic or geomorphological accuracy. For example, the piping phenomena are predominantly characterized by journalists as sinkhole formation due to their insufficient scientific background in geoscience. This may imply that, despite the news's significant potential and benefits for comprehending natural hazards, it is also necessary to carefully consider the news' scientific underpinnings.

**Conclusion**

It is essential to have a comprehensive and long-term understanding of how, when, and where natural hazards have affected societies in recent years to inform policymakers about how to overcome and mitigate multiple hazards. Therefore, with this study, we developed an approach to build spatiotemporal inventories from online gazette news for multiple hazards; flood, landslide, wildfire, and sinkhole formation by combining the web-scraping, semantic analysis, clustering, and geolocating algorithms on a national scale of Türkiye. The news parsing tool "tr-news-scraper" has been developed and 15569 news articles have been fetched with this tool from the selected online gazettes in Türkiye by employing the keywords associated with natural hazards. After NLP processing; a total of 13940 incidents of natural hazards have been recorded and geolocated. Consequently, we mapped 9609 floods, 1834 wildfires, 1843 landslides, and 654 sinkhole formation incidents, that occurred during the period between 1997 to 2023 in Türkiye. Our inventories show spatiotemporally distinct patterns in flood, landslide, wildfire, and sinkhole events, consistent with previous studies. Although the clustering and incident identification findings show 0.80 confidence scores, the contextual similarities, the contextual similarities (e.g., "yağış" term for flood and landslide), review news, and misinterpretations of the news may give rise to confusion either in fetching or clustering appropriate categories of natural hazards.

Overall, the approach provided in this study expands to existing inventories by investigating the potential and limitations of using web scraping, NLP, and machine learning methods, as well as providing an open alternative to creating inventories where others are inaccessible owing to national restrictions. Furthermore, we can more accurately portray natural hazard





events with these inventories because local news is less prevalent than global news but covers more events. Hence, further research is required to expand the spatial scale of similar approaches to other regions and multiple languages using advanced large language models.

## Data Availability

The data that support the findings of this study are available from the corresponding author upon reasonable request.

## Author contributions

AA and OD designed the study together with the contributions from TG. OD and AA performed data analyses, data visualization, and interpretations. AA prepared the manuscript with contributions from all co-authors.

## Financial Support

This study is supported by the 2247-A National Fellowship for Outstanding Researchers Program of the Scientific and Technological Research Council of Turkey (TUBITAK) [grant number 1199B472343092].

## Competing interests

The contact author has declared that none of the authors has any competing interests.

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
