# Peer review of "An automated approach for developing geohazard inventories using news: Integrating NLP, machine learning, and mapping."

_EGUsphere, 2025_

## Author Response (AR1)

**Response to Reviewers**

We appreciate your insightful and helpful remarks. After carefully reviewing every feedback and suggestion, we made the necessary revisions to the manuscript. An overview of the main modifications performed is provided below:

1) The clarification and interpretation of the related equation and figure 2 have been added to the manuscript.
2) A thorough description of uncertainty is provided below, and a new part describing newly conducted uncertainty analysis on location validations has been included in section 3.1.

We reply sentence to sentence below showing our thorough answers to each comment. Please see our answers given in bold style.

We have made an effort to address any concerns expressed while preserving the manuscript's clarity and scientific integrity. We would be pleased to respond as soon as possible to any further remarks or requests for clarification.

Regards,

**Reviewer #1**

I have read the manuscript entitled: An automated approach for developing geohazard inventories using news: Integrating NLP, machine learning, and mapping. Overall I think it is an interesting work whose objective is to develop an integrated method that recovers, classifies and geolocalizes multiple natural disasters. The basis of its study is the information published in some newspaper articles about natural hazards from Internet sources. The outline of the work is very well described in Figure 1.

**Response:** We thank the reviewer for valuable feedback. We have divided the reviewer's comments into relevant parts to better answer the raised points.

However, the manuscript is not clear in the following aspects: In section 2.2 the authors should clarify: The process of filtering the data or information obtained from newspaper sources. Equation 1 represents the probability of finding a term in the text, but Equation 2 does not have a clear interpretation. It is not explained why the logarithm is used in Equation 2.c The meaning of lines 170 and 171 is not clear.

**Response:** In our approach, we implement logarithmic scaling to address the frequency distribution of words within the dataset, as elucidated in the foundational equations. For example, ubiquitous terms such as "and" or "or" are found in nearly all 1,000 news articles, whereas a more specific term like "wildfire" may only appear in 10 articles. Absent logarithmic scaling, the term frequency ratio would be 1 for prevalent words (1,000/1,000) and 100 for less frequent words (1,000/10). Since similarity algorithms continue to prioritize these values, the lack of scaling

would result in an excessive focus on infrequent words. By employing a logarithmic transformation, we mitigate the influence of excessively common words that lack utility for classification, while preserving significant distinctions among pertinent terms. By making this change, the model is guaranteed to concentrate on informative terms instead of giving common ones an excessive amount of weight. We clarified the interpretation of Equation 2 in the manuscript by considering our response here. ***Please see the lines between 168 and 173.***

We have also clarified lines regarding the n-grams within the TfidfVectorizer by adding the explanation: *"By doing so, we benefit from more contextual meaning, for example, by maintaining word sequences, enabling models to differentiate phrases such as "sinkhole occurred" from the individual terms "sinkhole" and "occurred," which may possess distinct meanings when analyzed in apart from one another."* ***Please see the lines between 178 and 181.***

Figure 2 requires a clearer and more extensive explanation. Finally, I believe that the authors omit an analysis of the uncertainty in their results. Under these considerations, I believe that the work should be extensively revised.

**Response:** We made a better explanation in the Figure 2 caption, "*The world clouds illustrate the most frequently seen words in the filtered for different geohazard news. The sizes of each word denote its relative frequency or significance within the dataset; larger words, such as "waters, of the sinkhole, due to landslide, of the fire" signify principal themes, whereas smaller words offer supplementary context pertaining to details of geohazards (for example "meters depth sinkhole") within the news. The color variations serve solely for visual differentiation without indicating any categorical distinctions. Additionally, the spatial location of the words was arbitrarily positioned and does not indicate geographic relation with geohazards."*

We thank the reviewer for raising relevant points regarding the uncertainty allowing us a better opportunity for clarification of the limitations of our study. As indicated by reviewer #2, we aimed to compare the spatiotemporal performance of our inventories with existing literature in Türkiye. However, accessible spatiotemporal geohazard inventories are limited or do not exist in Türkiye hindering our capabilities in evaluating our inventory performance. As we made in "**Results and Discussion**" section, we compared our inventories with related but limited case studies, for example, landslides taking place in a particular region and time. Therefore, have decided to make an additional analysis, the ground truth evaluation step which we targeted to manually verify our approach. This method is followed by also related studies, extracting location from text-based data (Madruga de Brito et al., 2025; Stein et al., 2024). Here, we used random sampling as a ground truth evaluation step with 500 geohazard incidents to assess mapping performance. The random sampling resulted in 284, 97, 76, and 43 incidents of flood, landslide, wildfire, and sinkhole, respectively. We have manually checked these incidents and evaluated them by cross-checking the location of mapped geohazards and news context where we extracted location information. Our criteria were to achieve mapping the geohazard incidents to the center of the smallest administrative units which is available in the context of news. The uncertainty assessment for mapping performance overall resulted in good performance which is 82.4 % of geohazards accurately were mapped.

**Reviewer #2**

We appreciate your insightful and helpful remarks. After carefully reviewing every feedback and suggestion, we made the necessary revisions to the manuscript. An overview of the main modifications performed is provided below:

1) The assessment of potential accuracy changes regarding finer geographic units has been evaluated in detail.
2) A thorough description of uncertainty is provided below, and a new part describing newly conducted uncertainty analysis on location validations has been included to section 3.1.
3) Minor modifications have been fixed.

We reply sentence to sentence below showing our thorough answers to each comment. Please see our answers given in bold style.

We have made an effort to address any concerns expressed while preserving the manuscript's clarity and scientific integrity. We would be pleased to respond as soon as possible to any further remarks or requests for clarification.

Regards,

The manuscript addresses a relevant topic and proposes an interesting workflow for constructing geohazards inventories using online newspapers and natural language processing. It is well organized.

**Response:** We thank the reviewer for her/his insightful comments and we're happy to answer and clarify the points raised by the reviewer.

I have few comments:
- To further strengthen the manuscript, I recommend including more detailed explanations of the methods used—especially how potential biases from the model might influence the final dataset. A key concern is the geolocation strategy: if the exact street name is available, it is mapped to that street, otherwise it defaults to the city, and/or region (using the city/region's center, presumably). This same principle seems to apply to city- and street-level data as well. While this approach may work if we want to see the geohazards' distribution in a broader geographic area (e.g., all of Turkey), the uncertainty likely

increases when examining finer geographic units. It would be helpful to clarify how this method might affect the accuracy and reliability of analyses at smaller scales.

**Response:** The authors thank the reviewer for recommending the points here, which we found also relevant to raise these topics both as a reply here and in the manuscript. As indicated, geolocation was one of the most challenging parts of this study since we relied on text-based information within the online gazettes. This is, firstly, because of the inhomogeneous context writing style by journalism which we can't access the always similar "administrative level" information (city, county, and village) and details of this information, such as street, neighborhood, roads, etc. Therefore, our essential target for this study was to map the geohazards within these administrative levels by geolocating incidents to the center of these places, existing in the Open Street Map. We follow this procedure since we are not necessarily targeting to map geohazards (particularly landslides and sinkholes) geomorphologically meaningful terrains. Therefore, in this study, our target is to find and map the geohazard, temporally, within the smallest administrative level by taking the center of the cities, villages, etc. However, this procedure reveals spatially more accurate flood inventories compared to others since almost every newsworthy flood news occurs within the urbanized area of these administrative units. Secondly, the finer resolution such as street, and road information causes some problems as indicated initially due to the inhomogeneous context present within the news. For example, most of the news does not include street or road information for landslides and floods. Wildfire incidents naturally occur in forested areas most of the time outside of urbanized (but not necessarily, it might occur within the small forest patches in the urbanized area), and we made optimizations by assigning the wildfire incidents to the closest forested areas by using land use and land cover maps. Here, the most relevant geohazard for finer geographic units is flood incidents since most of the time floods occur in urbanized areas. However, since we are not able to extract – most of the time – specific street or road information, we prefer to geolocalize our inventories to the center of the administrative units. On the one hand, the problem for finer resolution, for example for street or road level, it is not possible to extract the information of which part, kilometers of this line-based location. On the other hand, since floods are represented by areal distribution, for example, inundation areas, further studies might provide better resolution by integrating a remote sensing-based approach to identify the exact location of these events within the urbanized area. Furthermore, achieving better accuracy or more accurate location representation for landslides and sinkholes requires also geomorphological interpretation, by integrating high-resolution satellite images to delineate their polygonal areas (particularly for landslides). We have clarified these issues by opening the section to the Results and Discussion, with the name "***Uncertainty assessment and limitations***".

- Although the study compares its overall results with established literature, a more granular validation (e.g., comparing known hazard events in specific cities or regions) would be needed. Consider adding such a validation step to illustrate both the strengths and limitations of the approach at different scales. Few diverse cases are sufficient.

**Response:** We are grateful that the reviewer agreed with our discussion points, in which we compared our findings with previous research. To answer the raised points by reviewers regarding the uncertainty assessment, which was also raised by reviewer #1, we opened a sub-section *"3.1 Uncertainty assessment and limitations"* in the ***Results and Discussion*** section. Here, we would like to primarily express that validation is obtained data is challenging in Türkiye, due to the lack of

complete and open-access inventories. Therefore section 3.1 mainly addresses and aims comparisons our inventories with the existing literature. However, most of these geohazard reports are case studies that investigate geohazards from a geological, geomorphological, or meteorological point of view rather than inventory assessment. Hence, to strengthen the reliability of our study, we have added the ground truth evaluation step which is a manual verification approach followed by also related studies (Madruga de Brito et al., 2025; Stein et al., 2024). Here, we used random sampling as a ground truth evaluation step with 500 geohazard incidents to assess mapping performance. The random sampling resulted in 284, 97, 76, and 43 incidents of flood, landslide, wildfire, and sinkhole, respectively. We have manually checked these incidents and evaluated them by cross-checking the location of mapped geohazards and news context where we extracted location information. Our criteria were to achieve mapping the geohazard incidents to the center of the smallest administrative units which is available in the context of news. The uncertainty assessment for mapping performance overall resulted in good performance which is 82.4 % of geohazards accurately were mapped.

- The first sentence in the Introduction states, "Natural disasters are vital. Could you please amend it?

**Response:** Yes, we replaced the first sentence with "*Geohazards are direct threats to human life, ecosystems, and societies worldwide socio-economically, demanding ongoing innovation and development in the mapping, analysis, and monitoring of these events.*"

- Up to the authors: Consider whether replacing natural hazards with the term geohazards.

**Response:** Thanks for the suggestion, we updated the terms natural hazards with geohazards to eliminate confusion that potentially might take place due to different terms for geohazards.

**References:**

Madruga de Brito, M., Sodoge, J., Kreibich, H., & Kuhlicke, C. (2025). Comprehensive Assessment of Flood Socioeconomic Impacts Through Text-Mining. *Water Resources Research*, *61*(1). https://doi.org/10.1029/2024WR037813

Stein, L., Mukkavilli, S. K., Pfitzmann, B. M., Staar, P. W. J., Ozturk, U., Berrospi, C., Brunschwiler, T., & Wagener, T. (2024). Wealth Over Woe: Global Biases in Hydro-Hazard Research. *Earth's Future*, *12*(10). https://doi.org/10.1029/2024EF004590

---

## Referee Report (RR1)

In this study, the authors developed a tool for the creation of inventories, which are of great importance in earth sciences and disaster studies in case from Turkey. On the other hand, they quickly created the locations of the created inventories and made them ready for use. This topic has a very important issue in disaster mitigation and modelling like machine learning techniques, especially for countries that do not have local data and have a very small piece in global data. I have read the article several times and I can say that its structure is well constructed and well written. However, I can say that there are a few minor points. I, therefore, recommend that the article be accepted after the following minor points have been dealt with.

Major Comments:

1) Here I would recommend that you give more emphasis to the generalization of the results of the study for use worldwide, especially in economically underdeveloped countries.

Minor Comments:

1) Figure 1 Raw News should be replaced with Unrefined News to be consistent with Table 2.

2) It would be better if you consider changing "Natural Hazard Inventory" to "Geohazard Inventory" since you use the "geohazard" in the manuscript.

3) The reason why NMF has been chosen might be added to the modeling section.

4) Open Street Map should be emphasized in the Geolocator section since you use the Nominatim tool.

5) It's up to the authors but consider replacing "online gazettes" with "newspaper" since you use mostly "newspaper" to to keep your manuscript consistent.

6) To demonstrate how the coherence score is regarded as an uncertainty indication, consider including a supporting sentence.

7) Consider changing the "research" with literature in this sentence "On the one hand, to enhance the reliability of our study, we incorporated a ground truth evaluation step, a manual verification method utilized in related research (Madruga de Brito et al., 2025; Stein et al., 2024)"

8) An explanation might be added to Figure 3 caption to clarify why the years vary in the X-axes of the plots.

9) "Yangın" (fire) and "orman" (forest) are the two most commonly (3.28% and 2.59%, respectively) used terms about wildfires." The details within the parenthesis should be added to the end of the sentence.

10) Can you better explain with a supportive sentence how you distinguished the urban fires?

---

## Author Response (AR2)

**Response to Reviewers**

We appreciate your insightful and helpful remarks. After carefully reviewing every feedback and suggestion, we made the necessary revisions to the manuscript. We reply sentence by sentence below, showing our thorough answers to each comment. Please see our answers given in bold style.

We have made an effort to address any concerns expressed while preserving the manuscript's clarity and scientific integrity. We would be pleased to respond as soon as possible to any further remarks or requests for clarification.

Regards,

**Reviewer #3**

In this study, the authors developed a tool for the creation of inventories, which are of great importance in earth sciences and disaster studies in case from Turkey. On the other hand, they quickly created the locations of the created inventories and made them ready for use. This topic has a very important issue in disaster mitigation and modelling like machine learning techniques, especially for countries that do not have local data and have a very small piece in global data. I have read the article several times and I can say that its structure is well constructed and well written. However, I can say that there are a few minor points. I, therefore, recommend that the article be accepted after the following minor points have been dealt with.

**Response:** We thank the reviewer for valuable feedback. We have divided the reviewer's comments into relevant parts to better answer the raised points.

Major Comments:
1) Here I would recommend that you give more emphasis to the generalization of the results of the study for use worldwide, especially in economically underdeveloped countries.

**Response:** We thank to comments of the reviewer and found it a reasonable point to highlight. Within the scope of this study, we first of all focused on the Türkiye case area to develop this automated approach for geohazard inventory development from the newsletters by integrating highly used, long-standing newsletters. It is because of the higher susceptibility of Türkiye to geohazards and available data sources. The generalization of the results of this study to global scale is also a valuable task for further studies by integrating large language models to eliminate the language barrier, which is a potential limitation hindering the generalization of this method. Therefore, we highlighted this point in the last sentence of the conclusion: *"Hence, further research is required to expand the spatial scale to generalize this study worldwide and across multiple languages by integrating advanced large language models. Please see the lines between 178 and 181*

Minor Comments:

1) Figure 1 Raw News should be replaced with Unrefined News to be consistent with Table 2.

**Response:** Thanks for this suggestion. We made this change.

2) It would be better if you consider changing "Natural Hazard Inventory" to "Geohazard Inventory" since you use the "geohazard" in the manuscript.

**Response:** We agree with the reviewer and have replaced it with Geohazard Inventory.

3) The reason why NMF has been chosen might be added to the modeling section.

**Response:** We have added the reason to the manuscript. *Please see the lines between 193 and 195*

4) Open Street Map should be emphasized in the Geolocator section since you use the Nominatim tool.

**Response:** We have added the Open Street Map to the Geolocator section. *Please see the line 234.*

5) It's up to the authors but consider replacing "online gazettes" with "newspaper" since you use mostly "newspaper" to to keep your manuscript consistent.

**Response:** We have replaced the online gazettes with the newspaper.

6) To demonstrate how the coherence score is regarded as an uncertainty indication, consider including a supporting sentence.

**Response:** A supportive sentence has been added. *Please see the lines between 311 and 313.*

7) Consider changing the "research" with literature in this sentence "On the one hand, to enhance the reliability of our study, we incorporated a ground truth evaluation step, a manual verification method utilized in related research (Madruga de Brito et al., 2025; Stein et al., 2024)"
**Response:** Change is done.

8) An explanation might be added to Figure 3 caption to clarify why the years vary in the X-axes of the plots.
**Response:** The clarification of the X-axes has been made in the caption.

9) "Yangın" (fire) and "orman" (forest) are the two most commonly (3.28% and 2.59%, respectively) used terms about wildfires." The details within the parenthesis should be added to the end of the sentence.

**Response:** The sentence has been updated. *"The most widely used phrases for wildfires are "yangın" (fire) and "orman" (forest), accounting for 3.28% and 2.59%, respectively."*

10) Can you better explain with a supportive sentence how you distinguished the urban fires?

 **Response:** The authors thank the reviewer for highlighting this point. We have explained this topic in the manuscript with an explanation of the selected keyword pairs. The bigram combinations in the selected news are the most important criteria to identify urban fires. For example, "orman yangını" which translates to wildfire, were the most important keyword pairs that we only took the news in our inventory. This procedure has been detailed in the Results and Discussion section. *Please see the lines between 257 and 265.*